# Antioxidant Properties and Prediction of Bioactive Peptides Produced from Flixweed (sophia, *Descurainis sophia* L.) and Camelina (*Camelina sativa* (L.) Crantz) Seed Meal: Integrated In Vitro and In Silico Studies

**DOI:** 10.3390/plants12203575

**Published:** 2023-10-14

**Authors:** Na Thi Ty Ngo, Tharindu R. L. Senadheera, Fereidoon Shahidi

**Affiliations:** Department of Biochemistry, Memorial University of Newfoundland, St. John’s, NL A1C 5S7, Canada; nttngo@mun.ca (N.T.T.N.); trlsenadheer@mun.ca (T.R.L.S.)

**Keywords:** bioactive peptides, bioinformatics, BIOPEP-UWM database, antioxidant activity, dipeptidyl peptidase IV (DPP IV) inhibitory peptide, angiotensin-converting enzyme (ACE) inhibitory peptides

## Abstract

Flixweed (sophia) seed meal and camelina, both by-products of oil processing, were employed to generate protein hydrolysates by applying Flavourzyme and Alcalase. This study aimed to integrate in vitro and in silico methods to analyze sophia and camelina protein hydrolysates for releasing potent antioxidative, dipeptidyl peptidase IV (DPP IV) inhibitors and angiotensin-converting enzyme (ACE) inhibitory peptides. In vitro methods were used to investigate the antioxidant potential of sophia/camelina protein hydrolysates. Bioinformatics techniques, including Peptideranker, BIOPEP, Toxinpred, AlgPred, and SwissADME, were employed to obtain the identification of bioactive peptides produced during the hydrolysis process. Protein hydrolysates produced from sophia and camelina seed meal exhibited higher ABTS and DPPH radical scavenging activities Ithan their protein isolates. Among the produced protein hydrolysates, Alcalase-treated samples showed the highest oxygen radical absorbance capacity and hydroxyl radical scavenging activity. In addition, sophia/camelina hydrolysates prevented hydroxyl and peroxyl radical-induced DNA scission and LDL cholesterol oxidation. In silico proteolysis was conducted on Alcalase-treated samples, and resultant peptides showed potential DPP IV and ACE-inhibitory activities. Identified peptides were further assessed for their toxicity and medicinal properties. Results indicate that all digestive-resistant peptides were non-toxic and had desirable drug-like properties. The findings of this study suggest that sophia/camelina protein hydrolysates are promising candidates for functional foods, nutraceuticals, and natural therapeutics.

## 1. Introduction

Natural plant-derived proteins, as opposed to those from animal sources, are currently attracting a lot of attention due to their sustainable availability and economic attraction [1]. Food-derived protein hydrolysates and their bioactive peptides are potentially active components suitable for formulating nutraceuticals and functional foods because they are safe and have a wide range of possible health benefits [2,3]. Thus, diversified sources of plant proteins increasingly address the global need for ingredients in the food industry. Furthermore, the identification of bioactive peptides encoded in oilseed processing waste has received much attention in recent years. It has also been highlighted as a viable biorefinery strategy for addressing the detrimental effects of unregulated waste disposal on the environment [3]. In this regard, camelina and sophia seed meals have emerged as potential functional food ingredients due to their nutritional value and beneficial health effects [4,5].

Protein hydrolysates are prepared by cleaving the peptide bonds, either enzymatically or chemically. The use of enzyme hydrolysis to manufacture protein hydrolysates has received much attention to create functional food ingredients and nutraceuticals for reducing disease risk and promoting health [6,7]. Many researchers have reported that peptides prepared by enzymatic hydrolysis from plant sources such as date seed, soybean, flaxseed, and quinoa exhibit a range of physiological and biological attributes, including antioxidative, dipeptidyl peptidase-IV (DPP IV) inhibition, and angiotensin-I-converting enzyme (ACE) inhibition [6,8,9,10]. According to the World Health Organization (WHO), the proportion of individuals affected by type 2 diabetes (T2D) and cardiovascular diseases (CVD) is growing tremendously. However, some synthetic therapeutic medications have been produced to treat these diseases, and many of them are considered harmful due to adverse effects such as inflammatory responses, taste disturbances, nausea, headaches, and allergic reactions [3,8]. Furthermore, oxidative damage is linked to several pathogenic problems, including cardiovascular disease, diabetes, cancer, and various chronic and inflammatory ailments [8]. For human disease prevention and management, natural antioxidants such as plant protein hydrolysates have arisen as an alternative to chemotherapy, as well as a strategy to maintain a healthy lifestyle.

Traditional methods have been employed for the discovery and creation of novel bioactive peptides, which include numerous processes for isolating, refining, and characterizing bioactive peptides. Nevertheless, this method is time-consuming and laborious. Currently, these problems have been solved by using bioinformatic methods that save time and work better than traditional methods, including the contributions of weakly polar interactions to structure and activity [11,12]. Generally, bioinformatics methods are designed to investigate bioactive peptides already present in protein sequences. This method may exclude potential precursor peptides as it only focuses on already studied protein sequences available in open-access databases. However, these virtual screening methods also detail the specificities and types of proteases used to generate bioactive peptide sequences [10]. In addition, the mass spectrometry (MS)-based proteomics method including sequence and conformation-specific structure determinations, a relatively recent technology, has been widely adopted and applied to the development of protein analysis to detect proteins and peptides from dietary sources [13,14]. After peptides are identified, the sequence can be further observed using bioinformatic tools, including UniProtKB, SwissProt, TrEMBL, BIOPEP, and PepBank, to assess the potential of proteins as sources for generating bioactive peptides [11,15]. A limited number of studies have reported the use of camelina and sophia meals as precursors for bioactive peptide. No study has so far been carried out to explore drug likeliness of camelina and sophia meal-derived bioactive peptides. This study marks the inaugural use of sophia and camelina seed meals for the creation of protein hydrolysates utilizing Flavourzyme and Alcalase, along with an examination of their antioxidant properties. This is also the first study to use in silico techniques to assess the antioxidative, ACE-inhibitory, and DPP IV-inhibitory activities of hydrolysates derived from camelina and sophia meals. The in silico techniques including PeptideRanker, BIOPEP, ToxinPred, and SwissADME, allergy FP tools have been used to screen these potential activities of camelina/sophia peptides to assess their toxicity and physicochemical characteristics, and drug likeliness were among variables tested. Therefore, the present study reports the importance of integrating in vitro and in silico techniques to identify the multifunctional peptides derived from underutilized camelina and sophia seed meals.

## 2. Results and Discussion

### 2.1. ABTS and DPPH Radical Scavenging Activities

The oxidative damage that occurs to the human body is caused by reactive oxygen species (ROS), namely superoxide anion (O_2_^•−^), peroxyl radicals (ROO^•^), hydroxyl radicals (HO^•^), hydrogen peroxide (H_2_O_2_), and peroxynitrite (ONOO). DPPH and ABTS are frequently employed for assessing the antioxidant potential of bioactive substances, including peptides and phenolic compounds [16,17,18,19,20]. 

Figure 1 and Figure 2 display the DPPH and ABTS radical scavenging activity of sophia/camelina hydrolysates and those of protein isolates; the hydrolysates exhibited significantly elevated values (*p* < 0.05) in comparison to their corresponding isolates. The significant difference observed between the hydrolyzed and non-hydrolyzed samples implies that sophia and camelina antioxidant peptides produced during enzyme digestion can donate electrons to decrease the content of ABTS^•+^. The ABTS radical scavenging activity of hydrolysates ranged between 26 and 46%, while DPPH radical scavenging activity varied from 17 to 36%. Among the tested samples, Alcalase-treated sophia protein isolate (SPI) exhibited the most notable (*p* < 0.05) capability in scavenging ABTS and DPPH radicals, whereas Flavourzyme hydrolyzed SPI showed the lowest radical scavenging activity. According to the previous study, the ability of food proteins and their hydrolysates to scavenge free radicals is influenced by various factors, including the amino acid composition, peptide sequences, and their size, as well as the protease’s specificity [21,22]. For example, aromatic amino acid residues (Tyr, His, Trp, and Phe) can donate hydrogen atoms to electron-deficient radicals through a procedure known as resonance stabilization. This enhances the radical-scavenging properties of the amino acid residues. Alkaline proteases, such as Alcalase and Flavourzyme, produce hydrolysates with more antioxidant activity compared to enzymes such as Neutrase, papain, bromelain, and pepsin [23]. This highlights that the observed trends in the radical scavenging abilities of sophia protein hydrolysates, particularly concerning DPPH and ABTS, are generated upon the treatment with Alcalase, which are comparable to those observed for camelina protein hydrolysates that were previously reported [21]. Therefore, our findings support the previously documented research into the radical scavenging potential of hydrolysates derived from diverse plant proteins [18,24].

### 2.2. Oxygen Radical Absorbance Capacity (ORAC) and Hydroxyl Radical Scavenging Activity 

The most damaging forms of oxidative stress are caused by hydroxyl radicals, which interact with involving biomolecules such as amino acids, DNA, proteins, and membrane lipids to cause severe cell damage [21]. Moreover, oxygen radicals play a major role in oxidative stress, resulting from an imbalance between reactive oxygen species (ROS) and antioxidants, and leading to fatal diseases such as cancer, cardiovascular diseases, arthritis, diabetes, and aging [6]. In the present study, the hydroxyl radical scavenging and oxygen radical absorbance of camelina and sophia protein hydrolysates are shown in Table 1. The ability to scavenge hydroxyl radicals was highest in sophia and camelina protein hydrolysates prepared with Alcalase around 3.13 and 3.21 (µm of Histidine/mg protein), respectively, and lowest in those prepared with Flavourzyme at 1.91 and 2.5 (µm of Histidine/mg protein). There was a range of values for the oxygen radical absorption capacity of camelina/sophia protein hydrolysates, from 0.51 to 1.13 (µm of trolox per mg of protein). The results indicated that the ability to scavenge hydroxyl radicals and oxygen radical absorbance capacity of camelina hydrolysates are higher than that of sophia hydrolysates. According to the findings of the research, hydrolysates that were generated from Alcalase gave the greatest value of hydroxyl radical scavenging ability as well as oxygen radical absorbance capacity. A significant difference (*p* < 0.05) existed in oxygen radical absorbance capacity and hydroxyl radical scavenging ability of sophia protein hydrolysates, which decreased in the order of AL > AL + FL > FL. Similar results for Alcalase-assisted hydrolyzed products of date seed meals and camelina meals have been reported in previous studies [6,21].

### 2.3. Metal Chelation Activity

The Fenton reaction, which involves lipid oxidation, leads to the degradation of hydroperoxide into volatile compounds. The chelation of transition metal ions (Fe, Cu, and Co) by hydrolysates reduces the intensity of the pink Ferrozine-ferrous complex [19]. The chelation of metal ions by sophia protein hydrolysates revealed significantly better metal-chelating activity than their respective protein isolates, as shown in Figure 3, and the findings regarding camelina hydrolysates have been published [21]. Furthermore, the chelation of ferrous ions by sophia protein hydrolysates varied greatly in the order of AL> AL + FL > FL. The highest metal chelation activity of sophia hydrolysates was observed in hydrolysates prepared by Alcalase, while AL + FL design produced the highest activity in camelina hydrolysates. This capacity correlates with the existence of amino acid residues such as tryptophan, aspartate, cysteine, histidine, and glutamate on the surface of proteins and polypeptides, which have been found to bind divalent metal ions. Additionally, because of the imidazole group that it possesses, the N-terminal presence of histidine demonstrates significant potential for metal ion chelation [25]. The findings suggest that peptides can interact with transition metal ions, potentially slowing down the oxidation process.

### 2.4. Inhibition against Copper-Induced LDL-Cholesterol Oxidation

LDL cholesterol plays a pivotal role in delivering triacylglycerols and cholesterol to body cells. The oxidation of plasma low-density lipoprotein (LDL), triggered by metal ions or reactive oxygen species (ROS), is a significant risk factor for atherosclerosis and coronary heart disease [26]. Cupric ion has been proven effective in initiating the oxidation of human LDL without EDTA, and the presence of cholesteryl linoleate and cholesteryl arachidonate in the LDL core supplies ample substrate for lipid peroxidation. According to the assay, Cu^2+^ is employed to induce in vitro human LDL-cholesterol oxidation, and the outcomes of conjugated dienes are measured at 234 nm. Carnosine is used as a positive control. Table 2 indicates that sophia, camelina hydrolysates, and carnosine displayed increased levels after 12 h of incubation, being 39.04−79.97 and 78.59%, respectively. Flavourzyme displayed the lowest amount of conjugated dienes production in comparison to all other treatments for protein hydrolysates. Camelina hydrolysates prepared by Alcalase have demonstrated their highest efficacy in inhibiting cupric ion-mediated LDL oxidation (79.97%). Camelina and sophia protein hydrolysates were found to have an inhibitory effect on human LDL cholesterol oxidation, which may be linked to their capacity to scavenge free radicals and chelate copper ions in LDL. In comparison to date seed protein, sophia/camelina protein hydrolysates were found to have a similar inhibitory impact on LDL oxidation [27].

### 2.5. Inhibition of Peroxyl and Hydroxyl Radical-Induced Supercoiled DNA Strand Scission 

Free radicals such as hydroxyl and peroxyl radicals generated in living cells possess a more significant reduction potential and can engage with biomolecules, leading to DNA damage at both the phosphate backbone and nucleotide bases. This can lead to mutation, carcinogenesis, and other pathological processes. As a result, limiting DNA oxidation is critical for maintaining cell health. In this experiment, AAPH, a water-soluble azo compound with cationic properties, served as the source of alkoxyl and alkyl peroxyl radicals. It was employed to induce single-strand cleavage of supercoiled plasmid DNA and evaluate the protective effects of hydrolysates against DNA oxidation. This was carried out to find out if hydrolysates have a protective effect against DNA oxidation. Carnosine was chosen as a positive control due to its natural antioxidant properties. The results are shown in Table 3. Sophia and camelina protein hydrolysates were found to reduce hydroxyl radical-induced DNA scission by 19.85 to 53.01% and 28.25 to 59.52%, respectively. Additionally, these hydrolysates exhibited a decrease in peroxyl radical-induced DNA oxidation by 62.83 to 87.10% and 75.82 to 88.77%, respectively. In both hydroxyl radical- and peroxyl radical-induced oxidation, hydrolysates that were produced by Alcalase and the combination of Alcalase and Flavourzyme showed a more potent inhibitory effect than carnosine. Comparable results were observed in the case of date seed protein hydrolysates, with supercoiled DNA retention ranging from 13 to 33% for hydroxyl radicals and from 47 to 83% for peroxyl radicals [27]. As a result, sophia/camelina protein hydrolysates have the potential to act as an appropriate dietary source for the protection of DNA against oxidation.

### 2.6. Amino Acid Composition

The amino acid content of protein hydrolysates has a major impact on their antioxidant activity and plays an important part in the physiological benefits that these substances provide [24]. Table 4 summarizes the amino acid composition of sophia protein hydrolysates prepared by Alcalase, and these correspond with those of Sophia seeds in a previous study [28]. Furthermore, the amino acid profile of camelina and sophia protein hydrolysates almost meets the recommendations of the Agriculture Organization and the World Health Organization/Food (FAO/WHO) for the amounts of most essential amino acids in food products. However, the levels of methionine and histidine in sophia hydrolysates are quite low compared to those recommended by WHO/FAO. In addition, the amounts of most amino acids in camelina hydrolysates are significantly higher than that of sophia hydrolysates. The amino acid profile of camelina and sophia protein hydrolysates exhibited a high concentration of leucine, tryptophan, arginine, glutamic, and aspartic. The high amounts of leucine found in hydrolysates may promote scavenging of superoxide radicals. Furthermore, negatively charged amino acids including Asp and Glu have strong antioxidant properties because they are rich sources of electrons that can be donated to free radicals, while tyrosine residue can act as a robust hydrogen donor, and histidine can be acknowledged for its effective radical scavenging activity, attributed to the breakdown of its imidazole ring [21,25]. As a result, the amino acid profile has been proposed as a method for predicting the antioxidant activity of protein hydrolysates. According to the findings of this study, camelina/sophia protein hydrolysates have the potential to serve as a valuable contributor of protein to food.

### 2.7. Peptide Ranker Analysis 

As shown in the research results decsribed above, the protein hydrolysate produced by Alcalase has high antioxidant activity. In this connection, peptideRanker (PepRank) is an in silico tool that has been used to predict the potential of bioactive peptides prepared using Alcalase. In general, any peptide with a threshold of more than 0.5 is considered bioactive. In addition, the false optimistic predictions reduce from 11 and 16% at a 0.5 threshold to 2 and 6% at a 0.8 threshold for long and short peptides, respectively. PepRank is more dependable that the peptide is bioactive when the expected probability is close to 1 [29]. Following the LC-MS/MS analysis, identified peptides from sophia/camelina protein hydrolysates were subjected to a peptide ranker tool to predict possible activity with a 0.8 threshold score. Table 5 shows 9 and 33 peptide sequences that were found to be potentially bioactive in sophia and camelina hydrolysates, respectively. These were with a threshold score of 0.8 to 0.96. FGFGPGL and GPPSGGGGGGGGGGGGGK peptides had the highest threshold score (0.96), while the lowest score was for SFPLPEL (0.8). Previous studies have shown that the biological activity of peptide sequence is determined by amino acid position and composition [30,31]. Therefore, the BIOPEP database was employed to investigate the antioxidative, ACE-inhibitory, and DPP IV-inhibitory effects of expected peptides [32].

### 2.8. In Silico Predictions of Potential Bioactive Peptides from Sophia/Camelina Protein Hydrolysate

Theprediction of the bioactive potential of selected peptides was demonstrated using the tool “Profiles of potential biological activity” from a BIOPEP-UWM database. This tool matches the predicted peptide sequences with antioxidative potential as well as ACE- and DPP IV-inhibitory activities to the available literature of diverse plant-, animal-, and other food-derived peptides [31,32]. Selected peptides were screened to investigate the type and locations of the bioactive fragments in peptide sequences. 

According to BIOPEP-UWM database, Table 6 indicate profiles of potential biological activity of camelina/sophia protein hydrolysates, including antioxidative, ACE- and DPP IV-inhibitory activities. A total of 5 of the 9 peptide sequences and 12 of the 33 peptides sequences released from sophia and camelina hydrolysates, respectively, were shown to have the potential to be antioxidative, and all selected sophia/camelina-derived bioactive peptides were found to have the ability to inhibit ACE and DPP IV activities (Table 6). All active fragments are dipeptides or tripeptides except for GFGPGL, which is easily absorbed in the gastrointestinal tract. The total number of ACE-inhibitory and DPP IV-inhibitory dipeptides is significantly higher than the total of tripeptides.

Interestingly, tryptophan (W) was the most common amino acid residue present in antioxidative peptides derived from sophia hydrolysates, such as FVPVTGLWM, WYTICICIL, and RAPWLEPL. In addition, GPP was the most common bioactive sequence identified in peptides derived from camelina hydrolysates. A similar result was found in antioxidant peptides from protein hydrolysate of bluefin leatherjacket [33]. In addition, the antioxidant activities of IY and IR were also predicted for jack bean and canavalin protein [34]. Peptides with more hydrophobic and aromatic amino acids were shown to have high antioxidant activity, and their presence is considered a critical attribute to the antioxidative property of bioactive peptides. In addition, the presence of hydrophobic amino acids including alanine (A), valine (V), leucine (L), isoleucine (I), proline (P), phenylalanine (F), tyrosine (Y), methionine (M), histidine (H), and tryptophan (W) is responsible for the antioxidant activity [35]. For instance, tyrosine and phenylalanine residues can act as potent hydrogen donors, and histidine can be credited with strong radical scavenging activity due to its imidazole ring [3]. Furthermore, the active amino acid residues located at the C-terminal as tyrosine in peptide AWPDKNPFFPSDPY, methionine in peptide GGGGGGGGPPAMSM, and GLDPPDLPM contributed to their antioxidant activity. Several studies have indicated that peptides having a tyrosine residue located at the C-terminus strongly scavenged hydroxyl radicals and peroxyl radicals as well as hydrogen peroxide [36]. As a result, the type and position of amino acid residues of the peptide could be comprehensively applied to identify novel antioxidant sources and obtain desirable antioxidants. Previous research has shown that several peptides found in sophia/camelina protein hydrolysates are potent antioxidants, and these peptides come from a variety of animal and plant sources. According to the database, AY, PW, LW, WY, IY, and IR were identified from okara protein, buckwheat, marine bivalve and bovine beta-lactoglobulin, jack bean, and canavalin protein, respectively. 

The highest number of potential ACE-inhibitory dipeptides and tripeptides (14) was found in the sequence SSTSGPAFNAGRSIWLPGWL. Numerous peptides found in sophia/camelina protein hydrolysates have previously been recognized as strong ACE inhibitors in diverse peptides from animal and plant origins. For example, GP, PL, GPL, and LPG were identified from Alaskan pollack skin, while CF, SF, GPP, and FVP were also previously confirmed from shark meat hydrolysate, garlic, wheat gliadin, and soya milk, respectively. Previous research has demonstrated that characteristics of peptides, such as chain length, composition, and sequence, can significantly affect their ACE-inhibitory activities. For example, the presence of hydrophobic amino acids such as proline (P), phenylalanine (F), isoleucine (I), valine (V), methionine (M), leucine (L), and alanine (A) close to the C-terminus of peptides promotes their binding to ACE and indeed inhibits the enzyme more effectively [6,37]. Additionally, the effect of proline on ACE-inhibitory activity is associated with its imidazole ring, which interacts strongly with the amino acid residues at the active centers of ACE. Proline was present in most ACE-inhibitory peptide sequences (Table 6). Thus, peptides derived from sophia/camelina meals are effective in inhibiting ACE. Alcalase was the most promising enzyme used for the isolation of both antioxidant and antihypertensive peptides, consistent with previous research on bioactive peptides derived from edible seeds [3]. 

According to Table 6, all the active segments are dipeptides or tripeptides, with dipeptides being the most abundant and effective DDP IV inhibitors. In addition, in active dipeptides and tripeptides, proline and alanine were the two most common amino acids such as GP, GA, PG, PA, AG, IP, LP, AP, PP, VP, APG, GPA, and PPL. This corresponds to some previous research results that peptides having proline or alanine residues in their sequences have potential DPP IV-inhibitory activity [38,39]. Furthermore, the potent DPP IV-inhibitory peptides have been found to possess a branched-chain amino acid (leucine, isoleucine, or valine) or an aromatic residue with a polar group in the sidechain (tryptophan) at their N-terminal position [40]. The interest in multifunctional peptides has been increasing as more of them have been discovered from food protein. According to the findings, it has been demonstrated that multifunctional peptides exhibit more than one major physiologically relevant bioactive characteristic. Because of this, there is more interest in multifunctional peptides now found in food proteins. Peptides, on the other hand, are susceptible to degradation during digestion in the gastrointestinal tract, and the biological activity of peptides can be either activated or inactivated depending on the structure and function of the peptides. Since in silico digestion is an appropriate method for predicting the release and resistance of bioactive peptides, the predicted peptides that were derived from the sophia/camelina protein hydrolysates were put through this process.

### 2.9. In Silico Simulated Gastrointestinal (GI) Digestion of Sophia/Camelina Protein Hydrolysates

Peptides that make up the building blocks of the human body are created when protein products are digested. To determine whether peptides can provide a health benefit, they must be subjected to simulated digestive conditions [2,41]. In this perspective, simulating GI digestion in silico is a valuable tool for determining the bioactive potential of sophia/camelina peptide before assessing its bioavailability and bioaccessibility in vivo. The sequences of protein hydrolysates were subjected to in silico proteolysis using pepsin (EC 3.4.23.1), trypsin (EC 3.4.21.4), and chymotrypsin (EC 3.4.21.1) for the prediction of theoretically released peptides by the enzymatic action program available in BIOPEP tool. In the present study, in silico proteolysis demonstrates the ability of given enzymes to release the antioxidative and the ACE-inhibitory peptides from sophia/camelina protein hydrolysates. 

After the simulated digestion process of sophia hydrolysates, none of the peptides displayed antioxidative potential, as predicted. This could be because there is insufficient information in the antioxidative peptide database, or it could be because gastrointestinal condition cleaves certain peptides [11]. The results showed that three out of the nine peptides identified from sophia hydrolysates were anticipated to release potent ACE-inhibitory and DPP IV-inhibitory fragments from their original sequences, respectively (Table 7). It is interesting to note that FGFGPGL possesses powerful ACE-inhibitory and DPP IV-inhibitory properties. GF is an active fragment sequence that can inhibit ACE as well as DPP IV. According to the findings given in Table 7, the bioactive potential upon GI digestion of peptides derived from camelina protein hydrolysate was reported. The results show that two peptides (AWPDKNPFFPSDPY, and SFPLPEL) have the potential antioxidant after the simulated digestion process. A total of 13 out of the 33 peptides identified from camelina hydrolysates were predicted to release potent ACE-inhibitory fragments from their original sequences. The ACE-inhibitory dipeptides are the most abundantly released from camelina protein hydrolysates, whereas few tri-peptides peptides sequences were found. Importantly, the frequency of release of potent ACE inhibitor peptides from each identified peptide is different because some peptides have more than one active fragment embedded in their sequence. For instance, SFPLPEL sequence was predicted to release three active ACE-inhibitory sequences, namely PL, SF, and PEL.

The “enzyme(s) action” option of the BIOPEP database is used to determine the degree of hydrolysis (DHt), the release frequency (AE), and the value of relative release frequency (W). A_E_ and W are the two most important parameters to consider when estimating the potential release of bioactive peptides from protein sequences [31]. In silico digestion revealed that sophia and camelina hydrolysates have degrees of hydrolysis ranging from 12.50 (DIPPPRGPL) to 66.67% (FHWDLPQ), as indicated in Table 8. The highest AE value (0.29) was observed in peptide SFPLPEL, whereas the lowest (0.06) was FGGYAPGILSPSPAML. Regarding camelina hydrolysates, the highest relative frequency of the release of fragments with a given selected enzymes (W) was 1.0 in AWPDKNPFFPSDPY, whereas AAMGGFPGGGGGAHALG (0.06) is the lowest value. After in silico simulated digestion of camelina meals derived peptides, most of the bioactive fragments possess proline (P), glycine (G), leucine (L), and phenylalanine (F) in their dipeptide and tripeptide sequences. The bioactive motifs that result contain VF, GPL, AW, DF, GL, GR, PPL, PQ, GGY, SF, DM, IL, GW, PL, and PEL. Previous studies using in silico methods to identify biopeptides in proteins from a variety of plants and animals have shown similar results [31,42]. According to the findings of several studies, the presence of hydrophobic amino acids (such as leucine, serine, and methionine) or aromatic amino acids (such as phenylalanine) in this peptide can boost the inhibition of ACE and DPP IV [41,43]. In addition, peptides that contain proline in their sequences have the potential to be highly effective ACE and DPP IV inhibitors [44]. The multiple bioactivities displayed by these peptides can increase their impact on ameliorating more than one disease target or various symptoms since many human diseases are interrelated in etiology and progress. According to these findings, sophia/camelina protein hydrolysates may be employed as multifunctional components in nutraceutical or functional food products.

### 2.10. Toxicity and Allergenicity Prediction of Sophia-Derived Bioactive Peptide Fractions after In Silico Digestion

The toxicity of bioactive peptides is a major concern when developing peptide-based nutraceuticals or functional food ingredients. The ability to predict the toxicity of therapeutic peptides prior to their synthesis is critical for reducing the time and money spent developing peptide-based drugs [45]. In addition, the majority of allergens are proteins present in a variety of plant and animal sources, resulting in a global problem that is continuously deteriorating [31]. As a result, all peptides derived from sophia/camelina protein hydrolysates following in silico digestion must be assessed for potential toxicity and allergenicity using the ToxinPred and allergen FP v.1.0 tools, respectively [31,46,47]. Table 9 illustrates that the low molecular weight dipeptides (GF, SL, GM, VF, AL, VL, AW, PF, PY, DF, GL, PM, GR, IH, QF, QW, PQ, SF, DM, IL, GW, and PL) and tripeptides (PGL, GPL, PPL, GGY, and PEL) that are released from sophia and camelina protein hydrolysates in silico are non-toxic. Non-toxic peptides are primarily composed of amino acids such as Val, Thr, Arg, Gln, Met, Leu, Lys, Ile, Phe, and Ala. These findings are comparable to those obtained with peptides derived from other protein sources, such as tubers and quinoa, rice bran, and flaxseed protein, all of which were found to be non-toxic [31,45,48,49].

According to Table 9, 4 out of 5 sophia-derived bioactive peptide fractions were classified as probable non-allergens, while 16 of 23 camelina-derived bioactive peptide fractions were classified as probable allergens and 6 peptides were probable non-allergens. Moreover, only VF is considered as non-allergen. The isoelectric points (pI) of the predicted bioactive peptides were found to be in the pH range of 3.80–10.11. In addition, 18 peptides have an isoelectric point of 5.88 with 0 net charge. The dipeptides (DF and DM) and tripeptides (PEL) showed a negative charge (−1) with a pI of 3.8 and 4.0, respectively, whereas GR had a positive charge (+1) with a pI of 10.11. Thus, bioactive peptides with an alkali isoelectric point had strong water solubility, whereas peptides with an acidic isoelectric point had poor water solubility. Some authors have investigated the charge, peptide sequence, low molecular weight, hydrophobicity, and hydrophilicity as important structural properties for food-derived peptide bioavailability [50,51]. In the present study, the active fragment fraction had molecular weights ranging from 188.25 Da (GL) to 357.44 Da (PEL). The hydrophobicity of the bioactive peptides was shown to be in the range of −0.38 (PQ) to 0.57 (VF), whereas their hydrophilicity varied from −2 (VF) to 0.85 (DM). As a result, these peptides have the potential to be used as functional ingredients in the food industry. Allergen and toxic effects of bioactive peptides can be predicted quickly and appropriately using in silico methods. However, in vitro and in vivo investigations are required to verify their toxicity and allergenicity to confirm the safety of these peptides for human consumption.

### 2.11. In Silico Evaluation of Drug-Likeness Bioactive Peptides Derived from Protein Hydrolysates

In recent years, ADME computation, which stands for absorption, distribution, metabolism, and excretion, has been very helpful in the in silico prediction of the drug-likeness of bioactive peptides. SwissADME is a web-based tool for predicting and assessing small molecule pharmacokinetics, drug-likeness, and medicinal chemistry friendliness [42,52]. The assessment of drug-likeness, which determines the prospect of a component becoming an oral drug, is one of the SwissADME evaluation methods. Because of its flexibility, this method is considered a cost-effective alternative to time-consuming experimental methods [53,54]. SwissADME was used to study physicochemical features such as the number of rotatable bonds (ROTB), hydrogen bond acceptors (HBA), hydrogen bond donors (HBD), topological polar surface area (TPSA), water solubility, lipophilicity, drug-likeness (Lipinski filter and bioavailability score), and pharmacokinetics (GI absorption). The drug-like characteristics of five multifunctional dipeptides and tripeptides that were released from sophia hydrolysates were used to predict the ADME and pharmacokinetic aspects of these compounds. The in silico physicochemical characteristics, drug-likeness, and pharmacokinetics of sophia peptides are displayed in Table 10 and Figure 4, respectively. All these parameters were examined with reference to captopril, which serves as the standard medication. Similarities could be seen between the drug-likeness and pharmacokinetics of the peptides and those of inhibitor drugs such as captopril. According to the data presented in Table 10, all the predicted dipeptides and tripeptides derived from sophia protein hydrolysates qualified as described by Lipinski. The Lipnski rule of five is considered as the rule of thumb to determine the drug likeliness of a molecule [53]. For instance, the rotatable bonds (RORB) of these peptides that were observed had a range of 0 to 9, anforheir TPSA values were lower than 130 Å^2^, which led to high gastrointestinal absorption. In addition, Table 10 shows all peptides (GF, SL, PM, M, and PGL) as being highly soluble and having high values for GI absorption. In terms of camelina hydrolysates, Table 10 reveals all peptides without PEL, rotatable bonds observed ranged from 0 to 9. This is consistent with previous studies because compounds with more than 10 rotatable bonds show poor oral bioavailability [52]. Furthermore, TPSA values of two dipeptides (GR, DM, and QW) and two tripeptides (GGY and PEL) are higher than 130 Å^2^ leading to low gastrointestinal absorption. Table 10 shows that all peptides were highly soluble and 17 out of 23 bioactive di- and tripeptide released from camelina hydrolysates had high values for GI absorption, except for GR, QW, PQ, DM, GGY, and PEL. In addition, all peptides (GF, SL, PM, M, and PGL) released from sophia protein hydrolysates were highly soluble and had high levels of GI absorption. These results showed Lipinski filter and bioavailability score of all the identified peptides were similar to that of captopril. This means that most of the peptides derived from camelina/sophia protein hydrolysates have very good absorption.

In addition, bioavailability as reflected in lipophilicity (LIPO), molecular size (Size), polarity (Polar), insolubility (Insolu), flexibility (Flex), and instauration (Insatu) were observed when radar was displayed for a quick assessment of drug-likeness. Oral bioavailability is best achieved in the area colored in Figure 4. According to Figure 4, all selected di- and tripeptides derived from sophia protein hydrolysates have their optimal values in the highlighted zone. In addition, the optimal values of most di- and tripeptides derived from camelina protein hydrolysates, except GR, DM, QW, GGY, and PEL, are in the colored region. As a result, peptides obtained from sophia and camelina protein hydrolysates have the potential to exhibit drug-like properties which could be exploited in the pharmaceutical industry. 

## 3. Materials and Methods 

### 3.1. Materials

In this study, camelina seeds were sourced from Linnaeus Plant Sciences INC in Saskatoon, SK, Canada, while sophia seeds were acquired from Daghdagh Abad, near Hamedan city in Iran, and were purchased at the Tavazo store in Toronto, ON, Canada. Flavourzyme (1000 LAPU/g) and Alcalase (2.4 AU/g) were procured from Novozymes in Bagsvaerd, Denmark. All chemical reagents used in the study were obtained from Fisher Scientific Ltd. in Ottawa, ON, Canada, or Sigma-Aldrich Canada Ltd. in Oakville, ON, Canada.

### 3.2. Methods

#### 3.2.1. Preparation of Sophia/Camelina Protein Isolatess

Sophia and camelina protein isolates were obtained from sophia and camelina seed meals using essentially the procedure outlined by Ngo and Shahidi [5] using defatted samples. The extracted proteins were freeze-dried and stored at −20 °C for further analyses.

#### 3.2.2. Preparation of Sophia/Camelina Protein Hydrolysates 

The hydrolysis of camelina and sophia protein isolates was performed using Alcalase (0.3 AU/g) and Flavourzyme (50 LAPU/g) under various conditions in a batch-wise manner [6,21]. In the case of enzyme combination treatments, the hydrolysis process began with Alcalase for 2 h (pH 8, 50 °C), followed by the addition of Flavourzyme for an additional 2 h (pH 7, 50 °C). The enzymatic reactions were terminated by heat inactivation at 90 °C for 10 min. The resulting hydrolysates were collected by centrifugation.

#### 3.2.3. DPPH Radical Scavenging Assay

The 2,2-diphenyl-1-picrylhydrazyl (DPPH) radical scavenging activity of both protein hydrolysates and their peptide fractions was assessed using a previously established method [16,55], with slight modifications as reported elsewhere using trolox as a reference [21]. The DPPH radical scavenging activity (%) was calculated in terms of micromoles (μM) of trolox equivalents (TE) per milligram of protein hydrolysate, using the following equation:(1)$$DPPH\;radical\;scavenging\;activity\;(\%)\;=\;\frac{\left(Absorbance\;of\;the\;control-Absorbance\;of\;the\;sample\right)}{Absorbance\;of\;the\;control}\times 100$$

#### 3.2.4. ABTS Radical Cation Scavenging Assay

The radical scavenging activity of protein hydrolysates was assessed using the 2,2′-azinobis (3-ethylbenzothiazoline-6-sulfonate) radical cation (ABTS•+) method with slight modifications as reported previously [21]. The ABTS radical scavenging activity was expressed in µmol of Trolox equivalents (TE) per milligram (mg) of protein hydrolysates, and calculated using the following equation:(2)$$ABTS\;radical\;scavenging\;ability\;(\%)\;=\;\frac{\left(Absorbance\;of\;the\;blank-Absorbance\;of\;the\;sample\;after\;6\;min\right)}{Absorbance\;of\;the\;blank}\times 100$$

#### 3.2.5. Hydroxyl Radical Scavenging Activity

The hydroxyl radical scavenging capacity was determined using EPR spectroscopy (Bruker E-scan, Bruker Biospin Co., Billericia, MA, USA) as reported by Ngo and Shahidi [21]. The hydroxyl radical scavenging capacity was quantified in terms of micromoles (μM) of histidine equivalents per milligram of protein hydrolysate.

#### 3.2.6. Oxygen Radical Absorbance Capacity (ORAC)

The ORAC assay was conducted utilizing a Fluostar Optima plate reader (BMG Labtech, Durham, NC, USA), following a previously described method with slight adjustments [6]. All chemicals and materials were diluted in a 75 mM/L phosphate buffer at pH 7.0 prior to the experiment. In Costar^®^ 3695 flat-bottomed 96-well black microplates (Corning Incorporated, Corning, NY, USA), camelina and sophia protein hydrolysates (1 mg/mL, 20 μL) were combined with 200 μL of fluorescein (0.11 μM in PBS). The mixture was pre-incubated for 15 min at 37 °C in the built-in incubator. The equipment was programmed to introduce 75 μL of AAPH (17.2 mg/mL in PBS) at the end of the incubation period during the first cycle. Changes in fluorescence were recorded every minute for 25 cycles, with each cycle lasting 210 s. Excitation and emission wavelengths were set at 485 nm and 520 nm, respectively. The ORAC values of protein hydrolysates were quantified as micromoles of trolox equivalents per milligram of protein using the trolox standard curve (ranging from 6.25 to 50 μM).

#### 3.2.7. Metal Chelation Activity

Ferrous ion chelating activity was assessed following a previously established method [6,21] using the trisodium salt of ethylenediaminetetraacetic acid (Na_3_EDTA) as a reference. The metal ion chelating ability (%) was calculated using the following formula:(3)$$Metal\;chelating\;activity\;(\%)\;=\;(1-\frac{Absorbance\;of\;the\;sample}{Absorbance\;of\;the\;control})\times 100$$

#### 3.2.8. Cupric Ion-Induced Human Low-Density Lipoprotein (LDL) Peroxidation

The inhibitory activity of camelina/sophia protein hydrolysates against copper-induced LDL cholesterol oxidation was measured according to the reported method [27]. LDL (5 mg/mL) was dialyzed in 10 mM phosphate buffer (pH 7.4, 0.15 M NaCl) using a dialysis tube with a molecular weight cut-off of 12−14 kDa (Fisher Scientific, Nepean, ON, Canada) at 4 °C under a nitrogen blanket in the dark for 12 h. Diluted LDL cholesterol (0.04 mg LDL/mL) was mixed with protein hydrolysate solutions (0.1 mg/mL). The positive control used was carnosine. The reaction was initiated by adding 0.1 mL of 100 μM solution of CuSO_4_. A blank containing only a sample without LDL or CuSO_4_ was prepared for each test compound. After incubation of the reaction mixture at 37 ℃ for 12 h, the conjugated dienes formed were recorded at 234 nm using a diode array spectrophotometer (Agilent, Palo Alto, CA, USA). The inhibitory effect of tested samples was expressed as percentage inhibition of conjugated diene formation according to the following equation:Inhibition (%) = [(A_control_ − A_sample_)/(A_control_ − A_native_)] × 100 (4)
A_control_ = (A°_0_ − A°_t_) (5)
A_sample_ = (A_0_ − A_t_) (6)
A_control_: the absorbance of LDL + CuSO_4_ + PBS(7)
A sample: the absorbance of LDL + CuSO_4_ + sample/standard(8)
A_native_: the absorbance of LDL + PBS.(9)

A_0_, A_t_, and A°_0_, A°_t_ are absorbance values for test samples and control, respectively, measured at zero time and at time t after incubation.

#### 3.2.9. Supercoiled DNA Strand Scission Inhibition 

The inhibitory activity, induced by peroxyl and hydroxyl radical, of protein hydrolysates from sophia/camelina isolates against DNA strand scission was determined according to previous studies, with some modifications [27]. The existing bands were visualised using the Alpha-Imager gel documentation system (Cell Biosciences, Santa Clara, CA, USA) under trans-illumination of UV light, and data processing was achieved using the Gel Analyzer 19.1 software. The percentage oxidation inhibition of the supercoiled DNA strand was calculated according to Equation (7).
Inhibition (%) = (DNA samples/DNA_blank_) × 100 (10)

#### 3.2.10. Amino Acid Composition

The amino acid composition analysis of each protein hydrolysate was conducted at the Analytics, Robotics, and Chemical Biology Centre (SPARC BioCentre), located at the Hospital for Sick Children in Toronto, ON, Canada, according to Mohan and Udenigwe [56]. All amino acids with the exception of tryptophan, cysteine, and methionine, were hydrolysed in 6 M HCl. Tryptophan, cysteine, and methionine analyses were carried out separately [21]. 

#### 3.2.11. Simulated Gastrointestinal Digestion

##### LC-MS/MS Analysis

The LC-MS/MS analysis was performed at the Analytics, Robotics, and Chemical Biology Centre (SPARC BioCentre) at The Hospital for Sick Children in Toronto, ON, Canada. This analysis utilized a Q_Exactive Orbitrap analyzer equipped with a nanospray source and an EASY-nLC nano-LC system from Thermo Fisher (San Jose, CA, USA), as previously described elsewhere [25]. Data analysis was performed using PEAKS+ studio 11 software (Bioinformatic Solutions, Waterloo, ON, Canada).

##### Peptide Ranking

PepRank (http://bioware.ucd.ie/~compass/biowareweb/ (accessed on 19 March 2022)) was used to screen peptides for their potential bioactivity which predicts the probability (between 0 and 1) of the peptide being bioactive. To limit the number of false positives, a threshold of 0.8 was selected [29]. The most potent camelina/ sophia-derived bioactive peptides were selected for further analysis.

##### Prediction of Potential Biological Activity Profile

The bioactive properties of selected peptides, including their antioxidant and antihypertensive activities, as well as their dipeptidyl peptidase IV inhibitory effects, were explored using the “profiles of potential biological activity” feature within the BIOPEP-UWM™ database, accessible at http://www.uwm.edu.pl/biochemia/index.php/en/biopep (accessed on 9 April 2022).

##### Prediction of the Bioactivities of the Released Peptides after Simulated Digestion

The identified sequences of bioactive peptides underwent in silico hydrolysis using the enzyme action functionality of BIOPEP-UWM™. This simulation employed pepsin (EC 3.4.23.1), trypsin (EC 3.4.21.4), and chymotrypsin (EC 3.4.21.1) as representative digestive enzymes to replicate the process of in vivo digestion. After that, the option to “search for active fragments” was chosen to get a list of peptides with potential activities such as antioxidant and antihypertensive activities and dipeptidyl peptidase IV inhibitory effects [14,52].

##### Toxicity and Allergenicity Prediction of Camelina/Sophia Bioactive Peptides Released after Simulated Digestion

One of the significant barriers to the sustainable utilization of camelina/ sophia proteins for the development of nutraceuticals and functional food ingredients is their toxicity and allergenicity. Therefore, the potential toxicity and allergenicity of the peptides released after simulated digestive proteolysis were investigated using ToxinPred online tool (https://webs.iiitd.edu.in/raghava/toxinpred/index.html (accessed on 25 April 2022)) [46] and allergen FP v.1.0 tool (http://www.ddg-pharmfac.net/AllergenFP (accessed on 25 April 2022)), respectively [57].

##### In Silico Physicochemical Properties and Drug-Likeness of Released Peptides after Simulated Digestion

The in silico assessment of the drug-like properties of the identified peptides derived from camelina and sophia was conducted, considering parameters related to absorption, distribution, metabolism, and excretion. This evaluation was carried out using the SwissADME tool, as previously outlined [53], and can be accessed at http://www.swissadme.ch/index.php# (accessed on 8 May 2022).

### 3.3. Statistical Analysis

All experiments were replicated three times, and the results were presented as the mean  ±  standard deviation. Statistical analysis was carried out using one-way ANOVA, and the means were compared using Tukey’s HSD test (*p*  <  0.05) with SPSS 16.0 for Windows (SPSS Inc., Chicago, IL, USA)

## 4. Conclusions

The current study explored the antioxidant potential of protein hydrolysates sourced from sophia and camelina using in vitro analysis. Based on in vitro results, Alcalase assisted protein hydrolysates were selected for further evaluation of their bioactive potential using peptidomics and bioinformatics tools. Findings indicate that protein hydrolysates produced from camelina and sophia seed meals could be utilized as precursors for releasing peptides with dual functions in ACE and DPP IV inhibition. Furthermore, the in silico predictions support the fact that these peptides were non-toxic and have drug-like properties. These findings revealed that camelina and sophia-derived peptides have the potential to serve as natural therapeutic ingredients as well as formulating functional food and nutraceuticals. However, future work should be focused on validating the optimal functionality of these bioactive peptides in vivo. 

## Figures and Tables

**Figure 1 plants-12-03575-f001:**
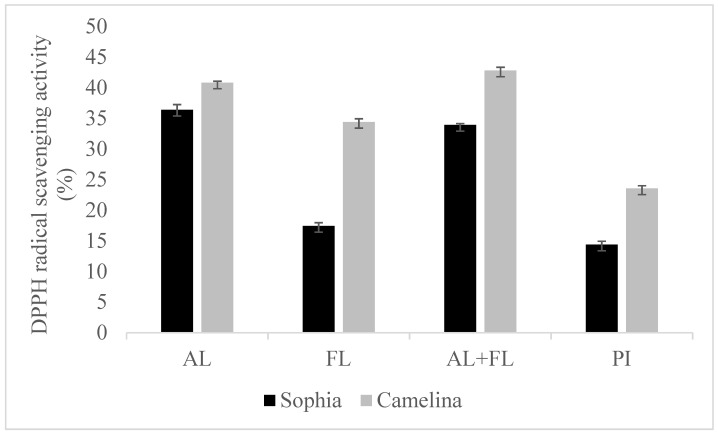
DPPH radical scavenging activity of sophia and camelina protein hydrolysates and their protein isolates (PI). FL, Flavourzyme; AL, Alcalase; and AL + FL, Combination of Alcalase and Flavourzyme.

**Figure 2 plants-12-03575-f002:**
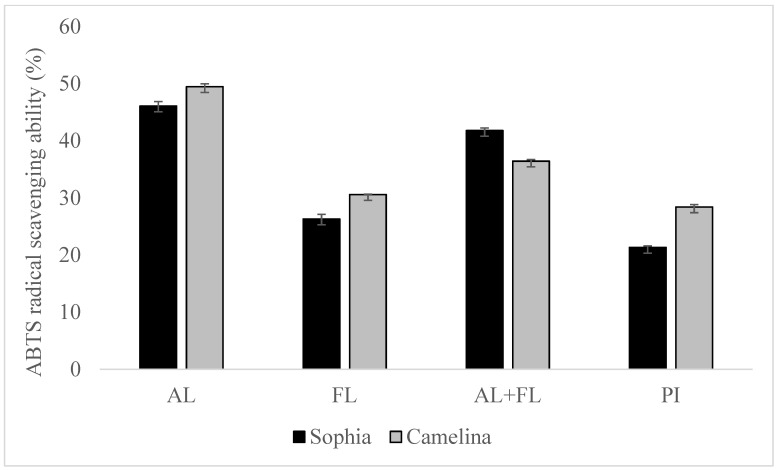
ABTS radical scavenging of sophia and camelina protein hydrolysates and their protein isolates (PI). FL, Flavourzyme; AL, Alcalase; and AL + FL, Combination of Alcalase and Flavourzyme.

**Figure 3 plants-12-03575-f003:**
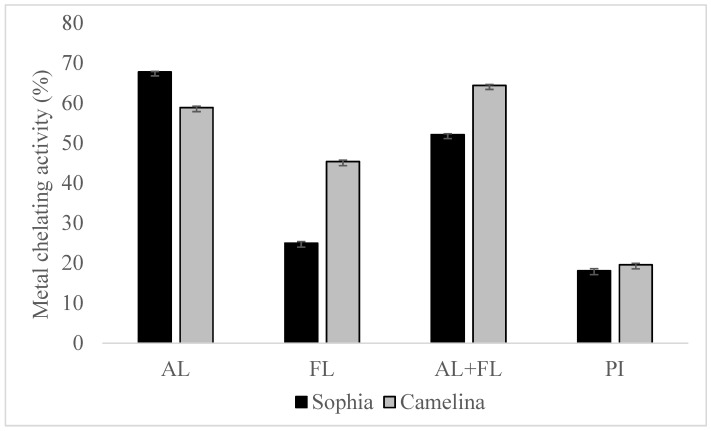
Metal chelation activity of sophia and camelina protein hydrolysates and their protein isolates (PI). FL, Flavourzyme; AL, Alcalase; and AL + FL, Combination of Alcalase and Flavourzyme.

**Figure 4 plants-12-03575-f004:**
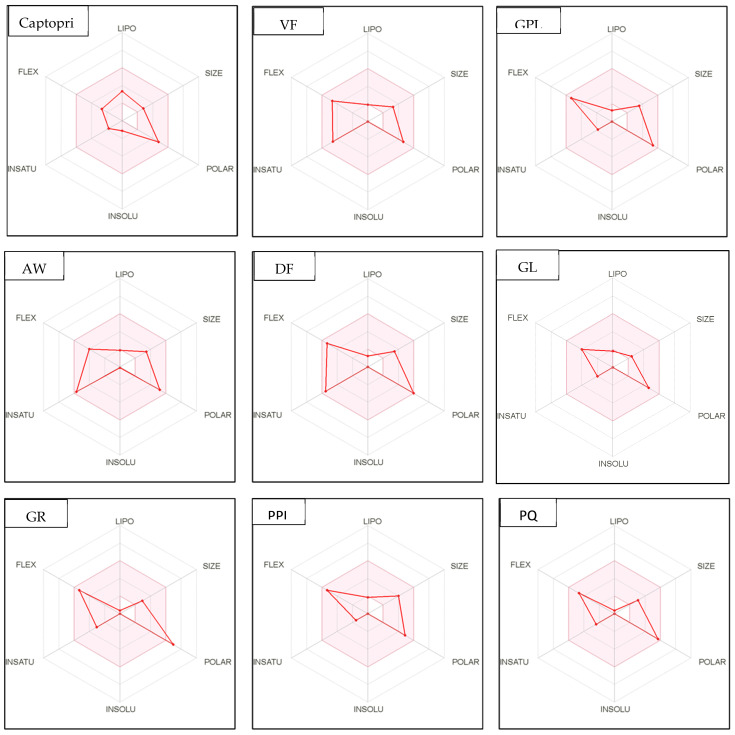
Bioavailability radar of sophia-derived bioactive peptides and camelina-derived bioactive peptides and inhibitor drug (captopril) based on physicochemical indices ideal for oral bioavailability. INSOLU, Insolubility: 0 < Log S (ESOL) < 6; INSATU, Instauration: 0.25 < Fraction Csp3 < 1; FLEX, Flexibility: 0 < Number of rotatable bonds < 9; LIPO, Lipophilicity: −0.7 < XLOGP3 < +5.0; SIZE, Molecular size: 150 g/mol < mol. wt. < 500 g/mol; POLAR, Polarity: 20 Å^2^ < TPSA <130 Å^2^. The colored region represents the appropriate physicochemical parameters for oral bioavailability.

**Table 1 plants-12-03575-t001:** Oxygen radical absorbance capacity (ORAC) and hydroxyl radical scavenging activity of camelina/sophia protein hydrolysates.

Samples	Protein Hydrolysates	Hydroxyl Radical Scavenging Activity	Oxygen Radical Absorbance Capacity
		(µm of Histidine/mg Protein)	(µm of Trolox/mg Protein)
Camelina	Alcalase	3.21 ± 0.03 a	1.13 ± 0.05 a
	Flavourzyme	2.50 ± 0.05 b	0.62 ± 0.04 b
	Alcalase + Flavourzyme	2.71 ± 0.16 c	0.93 ± 0.03 c
Sophia	Alcalase	3.13 ± 0.08 a	1.03 ± 0.05 a
	Flavourzyme	1.91 ± 0.05 b	0.51 ± 0.01 b
	Alcalase + Flavourzyme	2.63 ± 0.16 c	0.93 ± 0.04 c

Data are presented as mean ± SD (*n* = 3). Results with identical lowercase letters in the column do not exhibit significant differences at *p* < 0.05.

**Table 2 plants-12-03575-t002:** The inhibition against human LDL cholesterol oxidation by carnosine and sophia/camelina protein hydrolysates.

Protein Hydrolysates	LDL Oxidation Inhibition (%)	
	Sophia	Camelina
Carnosine	78.59 ± 0.11 a	78.59 ± 0.11 a
Alcalase	73.66 ± 1.19 b	79.97 ± 0.29 b
Flavourzyme	39.04 ± 0.28 c	45.94 ± 0.44 c
Alcalase + Flavourzyme	72.39 ± 0.59 b	62.68 ± 0.87 d

Data are presented as mean ± SD (*n* = 3). Results with identical lowercase letters in the column do not exhibit significant differences at *p* < 0.05.

**Table 3 plants-12-03575-t003:** Inhibition of peroxyl and hydroxyl radical-induced supercoiled DNA strand scission by sophia/camelina protein hydrolysates.

Samples	Protein Hydrolysates	DNA Scission Inhibition (%)	
		Hydroxyl Radical	Peroxyl Radical
Sophia	Carnosine	30.44 ± 1.36 d	22.63 ± 1.15 d
	Alcalase	53.01 ± 0.50 a	87.10 ± 2.35 a
	Flavourzyme	19.85 ± 0.87 b	62.83 ± 1.37 b
	Alcalase + Flavourzyme	46.35 ± 0.39 c	86.52 ± 0.32 a
Camelina	Alcalase	59.62 ± 2.21 a	88.77 ± 0.75 a
	Flavourzyme	28.25 ± 0.73 b	75.82 ± 1.55 b
	Alcalase + Flavourzyme	47.02 ± 0.35 c	80.07 ± 1.36 c

Data are presented as mean ± SD (*n* = 3). Results with identical lowercase letters in the column do not exhibit significant differences at *p* < 0.05.

**Table 4 plants-12-03575-t004:** Amino acid composition of sophia/camelina protein hydrolysates (Alcalase) (g/100 g).

Amino Acid	Sophia Hydrolysates	Camelina Hydrolyses [9]	WHO/FAO
**Essential amino acid (EAA)**			
Histidine (His)	0.9	2.1	1.6
Isoleucine (Ile)	1.4	2.9	1.3
Leucine (Leu)	2.2	4.9	1.9
Lysine (Lys)	1.4	2.3	1.6
Methionine (Met)	0.6	1.5	1.7
Phenylalanine (Phe)	1.5	3.2	
Threonine (Thr)	1.3	2.8	0.9
Tryptophan (Trp)	2.1	1.3	0.5
Valine (Val)	1.6	3.8	1.3
**Non-EAA (NEEA)**			
Alanine (Ala)	1.1	2.8	
Arginine (Arg)	2.8	7.0	
Aspartic acid +Asparagine (Asp + Asn)	3.0	6.4	
Cystine (Cys)	0.8	1.4	
Glutamic acid + Glutamine (Glu + Gln)	5.8	12.7	
Glycine (Gly)	1.8	3.2	
Proline (Pro)	1.5	3.4	
Serine (Ser)	1.3	3.0	
Tyrosine (Tyr)	1.2	2.4	
Phenylalanine + Tyrosine (Phe + Tyr)	2.7	3.3	1.9

**Table 5 plants-12-03575-t005:** Ranking of the predicted bioactive peptide sequences prepared sophia and camelina protein obtained by PepRank.

Samples	PepRank	Peptide Sequence
Sophia hydrolysates	0.96	FGFGPGL
	0.89	SSTSGPAFNAGRSIWLPGWL
	0.85	CAYDVAPGGLL
	0.85	SLCGIPPL
	0.83	PMITGFM
	0.82	FVPVTGLWM
	0.82	WYTICICIL
	0.82	RAPWLEPL
	0.81	LGMLPGL
Camelina hydrolysates	0.96	GPPSGGGGGGGGGGGGGK
	0.94	IDLFFVFL
	0.94	AAMGGFPGGGGGAHALGVL
	0.92	LNPCFTGGPLM
	0.92	NGGGGGGGGGGGPPKMVL
	0.91	PPPPGAL
	0.9	GGGGGGGFGGGAGGGLGGGGGL
	0.9	GGSPGIGGGL
	0.89	DIPPPRGPL
	0.89	GGGIGGGGGGGGGFGGGSGSGGGAGGGAGGGL
	0.89	LLGNGIGSGGGHGGKGGRVCY
	0.88	FGGGNLPAFVL
	0.88	AWPDKNPFFPSDPY
	0.88	FVPPFNPY
	0.86	DFSIFSPL
	0.85	FGGGNIPAFVL
	0.84	GGGGGGGGPPAMSM
	0.84	GLDPPDLPM
	0.83	VIGPGLGRDPFLL
	0.83	GGGAGGGGGL
	0.83	VFGSGLLGAFL
	0.83	FNAPIYL
	0.83	GPFGVIRPPL
	0.82	IHPIPPL
	0.82	AGAATGGFL
	0.82	NSGGGGGGWKGGGGQGGGWKGGGGQ
	0.81	QFQWIEFK
	0.81	FHWDLPQ
	0.81	FGGYAPGILSPSPAML
	0.81	FSFSPTVFDMILK
	0.81	FGWDKDL
	0.8	SFPLPEL

**Table 6 plants-12-03575-t006:** In silico predictions of potential bioactive peptides from sophia and camelina protein hydrolysate.

Samples	Peptide Sequence	Bioactive Segment Sequence		
		Antioxidative	ACE-Inhibitory Peptides	DPP IV-Inhibitory Peptides
		Peptides		
Sophia hydrolysates	FGFGPGL	GFGPGL	PGL, GP, GF, GL, FG, PG	GP, GL, GF, PG
	SSTSGPAFNAGRSIWLPGWL	-	GPA, LPG, P, IW, GW, AF, AG, GR, SG, PG, WL, SGP, ST, LP	GP, PA, LP, GPA, WL, AF, AG, FN, GW, IW, NA, PG, SI, TS
	CAYDVAPGGLL	AY	VAP, AY, AP, GL, GG, PG	VA, AP, LL, APG, GL, AY, GG, PG, YD
	SLCGIPPL	-	IPP, PL, IP, GI, PP, PPL	PP, IP, SL, PL, PPL, GI
	PMITGFM	-	GF, TG	GF, MI, PM, TG
	FVPVTGLWM	LWM, LW	LW, VP, GL, TG	VP, GL, WM, LW, PV, TG, VT
	WYTICICIL	WY, WYT	FVP, WM	WY, YT, IL, TI
	RAPWLEPL	PWL, PW	PL, AP, RA, WL	AP, RA, WP, PL, WL, PW
	LGMLPGL	-	PGL, LPG, GM, GL, LG, PG, LP	LP, GL, ML, PG
Camelina hydrolysates	GPPSGGGGGGGGGGGGGK	GPP	GP, GK, GG, SG, GPP, PP	GP, PP, GG, PS
	IDLFFVFL	-	VF, LF, FF	FL, VF, FF
	AAMGGFPGGGGGAHALGVL	AH, GAH	FP, AA, GF, GA, GV, MG, GG, LG,	HA, FP, GA, AL, AA, AH, GF, GG,
			PG, AH, LGV	GV, MG, PG
	LNPCFTGGPLM	-	LNP, GPL, GP, PL, GG, TG, CF, LN	GP, NP, PL, GG, LM, LN, TG
	NGGGGGGGGGGGPPKMVL	GPP	GP, GG, NG, PPK, GPP, PP	GP, PP, GG, MV, NG, PK, VL
	PPPPGAL	-	GA, PG, PP, PPP	PPPP, PP, GA, AL, PPG, PG
	GGGGGGGFGGGAGGGLGGGGGL	-	GF, GA, GL, AG, FG, GG, LG, FGG	GA, GL, AG, GF, GG
	GGSPGIGGGL	-	IG, GI, GL, GS, GG, PG	SP, GL, GG, GI, PG
	DIPPPRGPL	-	IPP, PR, GPL, GP, PL, IP, PP, PPP, RG	GP, PP, IP, PL, RG
	GGGIGGGGGGGGGFGGGSGS	-	GF, IG, GA, GL, AG, FG, GS,	GA, GL, AG, GF, GG, GI
	-GGGAGGGAGGGL		GG, SG, FGG	
	LLGNGIGSGGGHGGKGGRVCY	-	IG, GI, GH, GR, KG, GS, GK, HG, GG,	LL, GG, GH, GI, KG, NG
			SG, LG, NG, GHG	
	FGGGNLPAFVL	-	AF, FG, GG, FGG, LP	PA, LP, AF, GG, NL, VL
	AWPDKNPFFPSDPY	AW	FP, AW, FF	WP, FP, NP, AW, DP, PF, PS, PY, FF
	FVPPFNPY	-	VPP, VP, PP, FVP	PP, VP, NP, FN, PF, PY
	DFSIFSPL	-	PL, IF, DF	SP, PL, SI
	FGGGNIPAFVL	-	IPA, IP, AF, FG, GG, FGG	PA, IPA, IP, AF, GG, VL
	GGGGGGGGPPAMSM	GPP	GP, GG, GPP, PP	GP, PP, PA, GG
	GLDPPDLPM	LPM	DLP, GL, PP, LP	PP, LP, GL, DP, PM
	VIGPGLGRDPFLL	-	PGL, GP, IG, GL, GR, LG, PG	GP, LL, FL, GL, DP, PF, PG, VI
	VFGSGLLGAFL	-	VF, AF, GA, GL, FG, GS, SG, LG, AFL	GA, GL, AG, GG
	FNAPIYL	IY	IY, YL, AP	AP, FN, NA, PI, YL
	GPFGVIRPPL	IR	GP, PL, IRP, RP, FG, GV, PP, RPP	GP, PP, RP, PL, PPL, GV, IR, PF, VI
	IHPIPPL	-	IPP, PL, IP, PP, HP, PPL	PP, IP, HP, PL, PPL, IH, PI
	AGAATGGFL	GAA	AA, GF, GA, AG, GG, TG	GA, FL, AA, AG, AT, GF, GG, TG
	NSGGGGGGWKGGGGQGG	-	GW, KG, GQ, GG, QG, SG	WK, GG, GW, KG, QG
	-GWKGGGGQ			
	QFQWIEFK	-	IE, FQ, EF	WI, FQ, QF, QW
	FHWDLPQ	-	DLP, PQ, LP	LP, HW, PQ, WD, LPQ
	FGGYAPGILSPSPAML	-	GGY, GY, LSP, AP, YA, GI, FG, GG,	AP, PA, APG, SP, GG, GI, GY, IL, ML,
			PG, IL, FGG	PG, PS, YA
	FSFSPTVFDMILK	LK	VF, SF, PT, DM, IP	SP, IL, MI, PT, SF, TV, VF
	FGWDKDL	KD	GW, FG	GW, WD
	SFPLPEL	EL, PEL	PLP, FP, PL, SF, LP	LP, FP, PL, SF
	SGLGGGQGIGGGSGTGM	-	IG, GI, GM, GL, GS, GQ, GT, GG, QG,	GL, GG, GI, QG, TG
			SG, LG, TG, GTG	

**Table 7 plants-12-03575-t007:** Remaining bioactive properties after in silico simulated gastrointestinal digestion of peptides prepared sophia and camelina protein hydrolysate.

Sample	Peptide	Results of Enzyme Action	Location of Released Peptides	Active Fragment Sequence	Location	Bioactivity of Identified Peptide
Sophia hydrolysates	FGFGPGL	F-GF-GPGL	[1-1], [2-3], [4-7]	GF	[2-3]	ACE inhibitor, DPP IV inhibitor
	SSTSGPAFNAGRSIWLPGWL	SSTSGPAF-N-AGR- SIW-L-PGW-L	[1-8], [9-9], [10-12], [13-15], [16-16], [17-19], [20-20]	-	-	-
	CAYDVAPGGLL	CAY-DVAPGGL-L	[1-3], [4-10], [11-11]	-	-	-
	SLCGIPPL	SL-CGIPPL	[1-2], [3-8]	SL	[1-2]	DPP IV inhibitor
	PMITGFM	PM-ITGF-M	[1-2], [3-6], [7-7]	PM	[1-2]	DPP IV inhibitor
	FVPVTGLWM	F-VPVTGL-W-M	[1-1], [2-7], [8-8], [9-9]	-	-	-
	WYTICICIL	W-Y-TICICIL	[1-1], [2-2], [3-9]	-	-	-
	RAPWLEPL	R-APW-L-EPL	[1-1], [2-4], [5-5], [6-8]	-	-	-
	LGMLPGL	L-GM-L-PGL	[1-1], [2-3], [4-4], [5-7]	PGL	[5-7]	ACE inhibitor
				GM	[2-3]	ACE inhibitor
Camelina hydrolysates	GPPSGGGGGGGGGGGGGK	-	-	-	-	-
	IDLFFVFL	IDL-F-F-VF-L	[1-3], [4-4], [5-5], [6-7], [8-8]	VF	[6-7]	ACE inhibitor; DPP IV inhibitor
	AAMGGFPGGGGGAHALGVL	AAM-GGF-PGGGGGAH-AL-GVL	[1-3], [4-6], [7-14], [15-16], [17-19]	AL	[15-16]	DPP IV inhibitor
	LNPCFTGGPLM	L-N-PCF-TGGPL-M	[1-1], [2-2], [3-5], [6-10], [11-11]	-	-	-
	NGGGGGGGGGGGPPKMVL	N-GGGGGGGGGGGPPK -M-VL	[1-1], [2-15], [16-16], [17-18]	VL	[17-18]	DPP IV inhibitor
	PPPPGAL	-	-	-	-	-
	GGGGGGGFGGGAGG-GLGGGGGL	GGGGGGGF-GGGAGGGL -GGGGGL	[1-8], [9-16], [17-22]	-	-	-
	GGSPGIGGGL	-	-	-	-	-
	DIPPPRGPL	DIPPPR-GPL	[1-6], [7-9]	GPL	[7-9]	ACE inhibitor
	GGGIGGGGGGGGGFGGGSG-SGGGAGGGAGGGL	GGGIGGGGGGGGGF -GGGSGSGGGAGGGAGGGL	[1-14], [15-32]	-	-	-
	LLGNGIGSGGGHGGKGGRVCY	L-L-GN-GIGSGGGH -GGK-GGR-VCY	[1-1], [2-2], [3-4], [5-12],	-	-	-
	FGGGNLPAFVL	F-GGGN-L-PAF-VL	[1-1], [2-5], [6-6], [7-9], [10-11]	VL	[10-11]	DPP IV inhibitor
	AWPDKNPFFPSDPY	AW-PDK-N-PF -F-PSDPY	[1-2], [3-5], [6-6], [7-8], [9-9], [10-14]	AW	[1-2]	ACE inhibitor; DPP IV inhibitor; antioxidative
				PF	[7-8]	DPP IV inhibitor
	FVPPFNPY	F-VPPF-N-PY	[1-1], [2-5], [6-6], [7-8]	PY	[7-8]	DPP IV inhibitor
	DFSIFSPL	DF-SIF-SPL	[1-2], [3-5], [6-8]	DF	[1-2]	ACE inhibitor
	FGGGNIPAFVL	F-GGGN-IPAF-VL	[1-1], [2-5], [6-9], [10-11]	VL	[10-11]	DPP IV inhibitor
	GGGGGGGGPPAMSM	GGGGGGGGPPAM-SM	[1-12], [13-14]	-	-	-
	GLDPPDLPM	GL-DPPDL-PM	[1-2], [3-7], [8-9]	GL	[1-2]	ACE inhibitor; DPP IV inhibitor
				PM	[8-9]	DPP IV inhibitor
	VIGPGLGRDPFLL	VIGPGL-GR-DPF-L-L	[1-6], [7-8], [9-11], [12-12], [13-13]	GR	[7-8]	ACE inhibitor
	GGGAGGGGGL	-	-	-	-	-
	VFGSGLLGAFL	VF-GSGL-L-GAF-L	[1-2], [3-6], [7-7], [8-10], [11-11]	VF	[1-2]	ACE inhibitor; DPP IV inhibitor
	FNAPIYL	F-N-APIY-L	1-1], [2-2], [3-6], [7-7]	-	-	-
	GPFGVIRPPL	GPF-GVIR-PPL	[1-3], [4-7], [8-10]	PPL	[8-10]	ACE inhibitor; DPP IV inhibitor
	IHPIPPL	IH-PIPPL	[1-2], [3-7]	IH	[1-2]	DPP IV inhibitor
	AGAATGGFL	AGAATGGF-L	[1-8], [9-9]	-	-	-
	NSGGGGGGWKGGGGQGG-GWKGGGGQ	N- SGGGGGGW-K -GGGGQGGGW-K-GGGGQ	[1-1], [2-9], [10-10], [11-19], [20-20], [21-25]	-	-	-
	QFQWIEFK	QF-QW-IEF-K	[1-2], [3-4], [5-7], [8-8]	QF	[1-2]	DPP IV inhibitor
				QW	[3-4]	DPP IV inhibitor
	FHWDLPQ	F-H-W-DL-PQ	[1-1], [2-2], [3-3], [4-5], [6-7]	PQ	[6-7]	ACE inhibitor; DPP IV inhibitor
	FGGYAPGILSPSPAML	F-GGY-APGIL-SPSPAM-L	[1-1], [2-4], [5-9], [10-15], [16-16]	GGY	[2-4]	ACE inhibitor
	FSFSPTVFDMILK	F-SF-SPTVF-DM-IL-K	1-1], [2-3], [4-8], [9-10], [11-12], [13-13]	SF	[2-3]	ACE inhibitor; DPP IV inhibitor
				DM	[9-10]	ACE inhibitor
				IL	[11-12]	ACE inhibitor; DPP IV inhibitor
	FGWDKDL	F-GW-DK-DL	[1-1], [2-3], [4-5], [6-7]	GW	[2-3]	ACE inhibitor; DPP IV inhibitor
	SFPLPEL	SF-PL-PEL	[1-2], [3-4], [5-7]	SF	[1-2]	ACE inhibitor; DPP IV inhibitor
				PL	[3-4]	ACE inhibitor; DPP IV inhibitor
				PEL	[5-7]	antioxidative

**Table 8 plants-12-03575-t008:** In silico proteolysis of sophia and camelina protein hydrolysate for release bioactive peptides sequences.

Sample	Peptide	Active Fragment Sequence	DHt (%)	A_E_	W	Activity
Sophia hydrolysates	FGFGPGL	GF	33.33	0.14	0.14	ACE inhibitor
				0.14	0.25	DPP IV inhibitor
	SLCGIPPL	SL	14.29	0.13	0.17	DPP IV inhibitor
	PMITGFM	PM	33.33	0.14	0.25	DPP IV inhibitor
	LGMLPGL	PGL, GM	50	0.29	0.29	ACE inhibitor
Camelina hydrolysates	IDLFFVFL	VF	57.14	0.13	0.33	ACE inhibitor
				0.13	0.33	DPP IV inhibitor
	AAMGGFPGGGGGAHALGVL	AL	22.22	0.05	0.06	DPP IV inhibitor
	NGGGGGGGGGGGPPKMVL	VL	17.65	0.06	0.06	DPP IV inhibitor
	DIPPPRGPL	GPL	12.5	0.11	0.1	ACE inhibitor
	FGGGNLPAFVL	VL	40	0.09	0.14	DPP IV inhibitor
	AWPDKNPFFPSDPY	AW, PF	38.46	0.07	0.33	ACE inhibitor
				0.14	0.22	DPP IV inhibitor
				0.07	1	Antioxidative
	FVPPFNPY	PY	42.85	0.13	0.17	DPP IV inhibitor
	DFSIFSPL	DF	28.57	0.13	0.33	ACE inhibitor
	FGGGNIPAFVL	VL	30	0.09	0.14	DPP IV inhibitor
	GLDPPDLPM	GL, PM	25	0.11	0.25	ACE inhibitor
				0.22	0.4	DPP IV inhibitor
	VIGPGLGRDPFLL	GR	33.33	0.08	0.14	ACE inhibitor
	VFGSGLLGAFL	VF	40	0.09	0.11	ACE inhibitor
				0.09	0.17	DPP IV inhibitor
	GPFGVIRPPL	PPL	22.22	0.1	0.1	ACE inhibitor
				0.1	0.11	DPP IV inhibitor
	IHPIPPL	IH	16.67	0.14	0.14	DPP IV inhibitor
	QFQWIEFK	QF, QW	42.86	0.25	0.5	DPP IV inhibitor
	FHWDLPQ	PQ	66.67	0.14	0.33	ACE inhibitor
				0.14	0.2	DPP IV inhibitor
	FGGYAPGILSPSPAML	GGY	26.67	0.06	0.09	ACE inhibitor
	FSFSPTVFDMILK	SF, DM, IL	41.67	0.23	0.6	ACE inhibitor
				0.15	0.29	DPP IV inhibitor
	FGWDKDL	GW	50	0.14	0.5	ACE inhibitor
				0.14	0.5	DPP IV inhibitor
	SFPLPEL	PL, SF, PEL	33.33	0.29	0.4	ACE inhibitor
				0.29	0.5	DPP IV inhibitor
				0.14	0.5	Antioxidative

**Table 9 plants-12-03575-t009:** Prediction of toxicity and allergenicity of the potential bioactive fragments of sophia and camelina protein hydrolysate obtained after the simulation in silico digestion.

Samples	Active Fragment	Hydrophobicity	Hydrophilicity	Charge	pI	Molecular	Toxin	Allergenicity
	Sequence					Weight (Da)	Prediction	Prediction
Sophia hydrolysates	GF	0.39	−1.25	0	5.88	222.26	Non-Toxin	Probable allergen
	SL	0.14	−0.75	0	5.88	218.27	Non-Toxin	Probable non-allergen
	PM	0.1	−0.65	0	5.88	246.34	Non-Toxin	Probable non-allergen
	GM	0.21	−0.65	0	5.88	206.28	Non-Toxin	Probable non-allergen
	PGL	0.21	−0.6	0	5.88	285.38	Non-Toxin	Probable non-allergen
Camelina hydrolysates	VF	0.57	−2	0	5.88	264.34	Non-Toxin	Non-allergen
	AL	0.39	−1.15	0	5.88	202.27	Non-Toxin	Probable allergen
	VL	0.54	−1.65	0	5.88	230.33	Non-Toxin	Probable non-allergen
	GPL	0.21	−0.6	0	5.88	285.38	Non-Toxin	Probable allergen
	AW	0.31	−1.95	0	5.88	275.32	Non-Toxin	Probable allergen
	PF	0.27	−1.25	0	5.88	262.32	Non-Toxin	Probable allergen
	PY	−0.03	−1.15	0	5.88	278.32	Non-Toxin	Probable allergen
	DF	−0.05	0.25	−1	3.8	280.29	Non-Toxin	Probable allergen
	GL	0.35	−0.9	0	5.88	188.25	Non-Toxin	Probable allergen
	PM	0.1	−0.65	0	5.88	246.34	Non-Toxin	Probable non-allergen
	GR	−0.8	1.5	1	10.11	231.27	Non-Toxin	Probable allergen
	PPL	0.13	−0.6	0	5.88	325.44	Non-Toxin	Probable allergen
	IH	0.16	−1.15	0.5	7.1	268.34	Non-Toxin	Probable allergen
	QF	−0.04	−1.15	0	5.88	293.34	Non-Toxin	Probable allergen
	QW	−0.16	−1.6	0	5.88	332.38	Non-Toxin	Probable allergen
	PQ	−0.38	0.1	0	5.88	243.28	Non-Toxin	Probable allergen
	GGY	0.11	−0.77	0	5.88	295.33	Non-Toxin	Probable non-allergen
	SF	0.17	−1.1	0	5.88	252.28	Non-Toxin	Probable non-allergen
	DM	−0.23	0.85	−1	3.8	264.31	Non-Toxin	Probable allergen
	IL	0.63	−1.8	0	5.88	244.36	Non-Toxin	Probable non-allergen
	GW	0.27	−1.7	0	5.88	261.3	Non-Toxin	Probable non-allergen
	PL	0.23	−0.9	0	5.88	228.31	Non-Toxin	Probable allergen
	PEL	−0.05	0.4	−1	4	357.44	Non-Toxin	Probable allergen

**Table 10 plants-12-03575-t010:** In silico physicochemical properties and absorption, distribution, metabolism, excretion (ADME) profile of sophia and camelina bioactive peptides.

Samples	Active Fragment Sequence	Physicochemical Properties					Lipophilicity	Drug Likeliness		Pharmacokinetics
		ROTB	HBA	HBD	TPSA (Å²)	ESOL	C LogP	Lipinski Filter	Bioavailability Score	GI Absorption
Sophia hydrolysates	Captopril	4	3	1	96.41	−1.14	0.62	Yes (0)	0.56	High
	GF	6	4	3	92.42	0.32	−0.24	Yes (0)	0.55	High
	SL	7	5	4	112.65	0.84	−0.86	Yes (0)	0.55	High
	PM	7	4	3	103.73	1.04	−0.32	Yes (0)	0.55	High
	GM	7	4	3	117.72	1.32	−0.82	Yes (0)	0.55	High
	PGL	9	5	4	107.53	0.91	−0.51	Yes (0)	0.55	High
Camelina hydrolysates	VF	7	4	3	92.42	0.11	0.6	Yes (0)	0.55	High
	AL	6	4	3	92.42	1.05	−0.41	Yes (0)	0.55	High
	VL	7	4	3	92.42	0.5	0.27	Yes (0)	0.55	High
	GPL	8	5	3	112.73	0.99	−0.6	Yes (0)	0.55	High
	AW	6	4	4	108.21	−0.09	0.11	Yes (0)	0.55	High
	PF	6	4	3	78.43	0	0.33	Yes (0)	0.55	High
	PY	6	5	4	98.66	0.14	0	Yes (0)	0.55	High
	DF	8	6	4	129.72	0.83	−0.59	Yes (0)	0.56	High
	GL	6	4	3	92.42	0.86	−0.49	Yes (0)	0.55	High
	PM	7	4	3	103.73	1.04	−0.32	Yes (0)	0.55	High
	GR	8	5	5	156.82	2.31	−2.16	Yes (0)	0.55	Low
	PPL	8	5	3	98.74	0.09	0.06	Yes (0)	0.55	High
	IH	8	5	4	121.1	0.2	−0.33	Yes (0)	0.55	High
	QF	9	5	4	134.51	1.01	−0.68	Yes (0)	0.55	High
	QW	9	5	3	151.3	0.49	−0.48	Yes (0)	0.55	Low
	PQ	7	5	4	121.52	2.19	−1.6	Yes (0)	0.55	Low
	GGY	9	6	5	141.75	1.14	−1.22	Yes (0)	0.55	Low
	SF	7	5	4	112.65	1.03	−0.77	Yes (0)	0.55	High
	DM	9	6	4	155.02	1.67	−1.2	Yes (0)	0.11	Low
	IL	8	4	3	92.42	0.25	0.63	Yes (0)	0.55	High
	GW	6	4	4	108.21	0.23	−0.21	Yes (0)	0.55	High
	PL	6	4	3	78.43	0.66	0.04	Yes (0)	0.55	High
	PEL	12	7	5	144.83	0.92	−0.52	Yes (0)	0.56	Low

Abbreviations: ROTB, number of rotatable bonds; HBA, number of hydrogen bond acceptors; HBD, number of hydrogen bond donors; TPSA, topological polar surface area; ESOL, estimate solubility; C LogP, logarithm of compound partition coefficient between n-octanol and water; GI absorption, gastrointestinal absorption.

## Data Availability

Please contact Dr. Fereidoon Shahidi for data requests.

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
