# Peer review of "Antioxidant Properties and Prediction of Bioactive Peptides Produced from Flixweed (sophia, Descurainis sophia L.) and Camelina (Camelina sativa (L.) Crantz) Seed Meal: Integrated In Vitro and In Silico Studies"

_plants, 2023, doi:10.3390/plants12203575_

Round 1
Reviewer 1 Report
The work, however interesting, is not very attractive. There are no new molecules. There is a list of experiments conducted on two sophia/camelina protein isolates, with different techniques. The results are interesting but more than a entire work it seems an account of experimental data. Several in silico studies are carried out but seem unrelated to the rest of the work. There is a lack of an overview, no conclusions. It is not clear what the overall objective of the work is.
Author Response
Response to Reviewer 1 Comments:
The work, however interesting, is not very attractive. There are no new molecules. There is a list of experiments conducted on two sophia/camelina protein isolates, with different techniques. The results are interesting but more than a entire work it seems an account of experimental data. Several in silico studies are carried out but seem unrelated to the rest of the work. There is a lack of an overview, no conclusions. It is not clear what the overall objective of the work is.
Response: Thank you for your comment on the manuscript. We have updated the abstract, introduction and conclusion sections highlighting the importance of using integrated in vitro and in silico methods. Our overall objective is to effectively evaluate the bioactive potential and physicochemical properties of peptides encrypted in camelina and sophia seed meal using both analytical and bioinformatic approaches. We used in vitro methods to identify most promising protein hydrolysates derived from camelina and sophia seed meals and then used that hydrolysate for further analysis its multifunctional properties using peptidomics and bioinformatics tools. Use of in silico approaches can offer a valuable alternative to expensive wet-lab methods and represent a convenient and efficient approach to identify the multifunctionality of the target peptides.
Response to Reviewer
Reviewer 2 Report
The manuscript presents the utilization of sophia and camelina protein hydrolysates for releasing potent antioxidative and dual DPP IV-ACE inhibitory peptides, and the subsequent property measurement and prediction of peptide properties using various in silico approaches.
First and foremost, the title should be modified to precisely reflect the content of the study.
Dissemination is okay, yet Figures 1 + 2 are not mentioned in the text. Further, there are no fewer than 10 tables in the main text which is far too many, with most of these able to be moved to an electronic supplementary information file, with a summary made in the text. The authors can make summary plots or similar graphics for these bus-schedule like lists. They need to be condensed and could be single-lined each; Table-1, ESOL column should be changed to requiring 1 and not 3 lines, but creating shorthand or acronym labels for some columns and re-sizing others.
Much of the work involves results-reporting, with no evolution to the results beyond what is directly in the graphics, tables or data. True interpretation and discussion are lacking and need evolving and disseminating in the main work.
With respect to referencing, it appears that references [17-27] are missing from the text. This referee fails to find them; most likely culprit is this old referee's old and failing eyesight to see these. Yet, the authors should double check and fix these as need-be. Likewise for the self-citation which sits at near-15 % (7/49), which is most problematic for the several of these that self-reference to very general statements or phenomena (see this for refs [4], [8], [9], [30]. As text is missing for references to [17], [19], [27] one cannot comment, yet once fixed the authors should ensure these references are necessarily not self-citing.
Further, the following sentences should be refined to reflect the contributions of well-established works using highly-accurate in silico methods, including electronic structure determinations that form strong agreement between experiment and modelling/computation:
"Currently, these problems have been solved by using bioinformatic methods that save time and work better than traditional methods [14], including the contributions of weakly-polar interactions to structure and activity[14b].
(Page-2, Introduction, 3rd paragraph, lines 65-66)
"Besides, mass spectrometry (MS)-based proteomics including sequence and conformation specific structure determinations[14c], a relatively recent technology, has been widely adopted and applied in the development of protein analysis ...
(Page-2, Introduction, 3rd paragraph, lines 65-66)
Suggested reference titles:
[14b] = The role of enhanced aromatic-electron donating aptitude of the tyrosyl sidechain with respect to that of phenylalanyl in intramolecular interactions.
[14c] = A Hartree–Fock, MP2 and DFT computational study of the structures and energies of ″b2 ions derived from deprotonated peptides. A comparison of method and basis set used on relative product stabilities.
The statement “Currently, these problems have been solved by using bioinformatic methods that save time and work better than traditional methods.” further needs to be reformulated to state the limitation of these methods, including precision, etc.
Please double check Table 2 Carnosine row.
It is not clear what “…as described by Lipinski” refers to.
The website URL should only be listed in the reference list, not in the text, and be accompanied by access date.
minor editing
Author Response
Response to Reviewer 2 Comments
The manuscript presents the utilization of sophia and camelina protein hydrolysates for releasing potent antioxidative and dual DPP IV-ACE inhibitory peptides, and the subsequent property measurement and prediction of peptide properties using various in silico approaches.
First and foremost, the title should be modified to precisely reflect the content of the study.
Response: Thank you for your comment on the manuscript. We have updated the title of the study.
Dissemination is okay, yet Figures 1 + 2 are not mentioned in the text. Further, there are no fewer than 10 tables in the main text which is far too many, with most of these able to be moved to an electronic supplementary information file, with a summary made in the text. The authors can make summary plots or similar graphics for these bus-schedule like lists. They need to be condensed and could be single-lined each; Table-10, ESOL column should be changed to requiring 1 and not 3 lines, but creating shorthand or acronym labels for some columns and re-sizing others.
Response: Thank you for your comment on the manuscript. The results of DPPH and ABTS radical scavenging have been shown in Figures 1 and 2, and their discussion is provided in section 2.1 (line 98). In terms of in silico approaches, the tables provided detailed information from each research. These tables were crucial for our discussions. While we couldn't condense them per the reviewer's recommendation, we've modified Table 10 and adjusted the formatting of Tables 5, 6, and 7.
Much of the work involves results-reporting, with no evolution to the results beyond what is directly in the graphics, tables or data. True interpretation and discussion are lacking and need evolving and disseminating in the main work
Response: Thank you for your comment on the manuscript. We have added some references and have clarified some discussions in the highlighted manuscript.
With respect to referencing, it appears that references [17-27] are missing from the text. This referee fails to find them; most likely culprit is this old referee's old and failing eyesight to see these. Yet, the authors should double check and fix these as need-be. Likewise for the self-citation which sits at near-15 % (7/49), which is most problematic for the several of these that self-reference to very general statements or phenomena (see this for refs [4], [8], [9], [30]. As text is missing for references to [17], [19], [27] one cannot comment, yet once fixed the authors should ensure these references are necessarily not self-citing.
Response: Thank you for this comment. We have checked and added some references. In addition, we used Endnote software to cite references to help reviewers check references more easily.
Further, the following sentences should be refined to reflect the contributions of well-established works using highly-accurate in silico methods, including electronic structure determinations that form strong agreement between experiment and modelling/computation:
"Currently, these problems have been solved by using bioinformatic methods that save time and work better than traditional methods [14], including the contributions of weakly-polar interactions to structure and activity[14b].
(Page-2, Introduction, 3rd paragraph, lines 65-66)
"Besides, mass spectrometry (MS)-based proteomics including sequence and conformation specific structure determinations[14c], a relatively recent technology, has been widely adopted and applied in the development of protein analysis ...
(Page-2, Introduction, 3rd paragraph, lines 65-66)
Suggested reference titles:
[14b] = The role of enhanced aromatic-electron donating aptitude of the tyrosyl sidechain with respect to that of phenylalanyl in intramolecular interactions.
[14c] = A Hartree–Fock, MP2 and DFT computational study of the structures and energies of ″b2 ions derived from deprotonated peptides. A comparison of method and basis set used on relative product stabilities.
Response: Thank you for these comments. We have included these two references and the sentences has been rewritten (lines 66-67; 72-73).
The statement “Currently, these problems have been solved by using bioinformatic methods that save time and work better than traditional methods.” further needs to be reformulated to state the limitation of these methods, including precision, etc.
Response: The sentence has been rewritten and included the limitations (lines 68-72).
Please double check Table 2 Carnosine row.
Response: Carnosine were used as a positive control for both camelina and sophia hydrolysates. We have updated Table 2.
It is not clear what “…as described by Lipinski” refers to
Response: New sentence was added to explain the Lipinski rule.
The website URL should only be listed in the reference list, not in the text, and be accompanied by access date
Response: Thank you for your comment. We have added the web site URL and the access date.
Reviewer 3 Report
Manuscript “Antioxidant Properties and Prediction of Bioactive Peptides Produced from Camelina (Camelina sativa (L.) Crantz) and flixweed (sophia, Descurainis sophia L.) Seed Meal” is a good and informative study but had some minor corrections. Below are some comments/suggestion for the authors to improve its quality:
- Language should be improvised.
- Manuscript contains 30% similarity with already published article. Reduce the % of plagiarism upto 10%
- In Abstract section methodology of study should be included.
- In Abstract, authors should write at least two sentences about the materials and methods which have been used for this review article and sources that they did use to search for articles.
- Also abstract section should include some conclusive statements.
- Introduction section must include the need and significance of study.
- Materials and Methods should be rewrite.
- Conclusions: The conclusions are too general, format according to future aspects. Please make them more specific.
- Carefully read whole manuscript line by line and improve the sentence formation
- Please, add more updated references about the topic in different sections.
- Cross check all references and style of reference according to Journal format, use abbreviation of journal name in reference.
The work is interesting and worth publishing.

- Manuscript contains 30% similarity with already published article. Reduce the % of plagiarism upto 10%
Author Response
Response to Reviewer 3 Comments
Manuscript “Antioxidant Properties and Prediction of Bioactive Peptides Produced from Camelina (Camelina sativa (L.) Crantz) and flixweed (sophia, Descurainis sophia L.) Seed Meal” is a good and informative study but had some minor corrections. Below are some comments/suggestion for the authors to improve its quality:
Response: Thank you for your suggestions.
- Language should be improvised.
Response: We have updated the abstract, introduction, method, results and discussion and the conclusion sections.
- Manuscript contains 30% similarity with already published article. Reduce the % of plagiarism upto 10%.
Response: Thank you for your comments. However, the manuscript had already passed the plagiarism check, and the comment is for something other than this article. We saw the attached file in your review; that is not on our manuscript.
- In Abstract section methodology of study should be included.
- In Abstract, authors should write at least two sentences about the materials and methods which have been used for this review article and sources that they did use to search for articles.
- Also abstract section should include some conclusive statements.
Response: Thank you for these suggestions. We have revised the abstract section.
- Introduction section must include the need and significance of study
Response: We have revised the introduction section and included the overall significance of the study (Lines 88-90).
- Materials and Methods should be rewrite.
Response: The same as comment 2, we revised the materials and methods.
- Conclusions: The conclusions are too general, format according to future aspects. Please make them more specific.
Response: Thank you for these suggestions. We have revised the conclusion section.
- Carefully read whole manuscript line by line and improve the sentence formation
- Please, add more updated references about the topic in different sections.
- Cross check all references and style of reference according to Journal format, use abbreviation of journal name in reference.
Response: Thank you for these suggestions. We have updated and checked all references.
Round 2
Reviewer 1 Report
Although I still consider the topic of the manuscript not of great interest to the scientific community, I think that this new version is clearer and fluent, certainly improved. With this new aspect, I would recommend its publication.
Reviewer 2 Report
All issues have been addressed.
There was a misunderstanding, url with access date should be listed as a reference in the Reference section, and in the text should be written as "PepRank [ref No. ]..."
Reviewer 3 Report
Accepted in present form